

**Will landscape responses reduce glacier sensitivity to climate change in High Mountain**
**Asia?**
Stephan Harrison[1], Adina Racoviteanu[2], Sarah Shannon[3], Darren Jones[1], Karen Anderson[4],
Neil Glasser[5], Jasper Knight[6], Anna Ranger[1], Arindan Mandal[7], Brahma Dutt Vishwakarma[7],
Jeffrey S. Kargel[8], Dan Shugar[9], Umesh Haritashya[10], Dongfeng Li[11], Aristeidis Koutroulis[12],
Klaus Wyser[13], Sam Inglis[14]
Affiliations
1 College of Environment, Science and Economy, Exeter University, Exeter, UK
2 Université Grenoble Alpes, CNRS, IRD, IGE – 38400 Saint Martin d'Hères, France
3 Bristol Glaciology Centre, Department of Geographical Science, University Road, University
of Bristol, BS8 1SS, UK
4 Environment and Sustainability Institute, University of Exeter, Penryn Campus, Penryn,
Cornwall, TR10 9EZ, UK
5 Centre for Glaciology, Department of Geography and Earth Sciences, Aberystwyth
University, Wales SY23 3DB, UK
6 School of Geography, Archaeology & Environmental Studies, University of the
Witwatersrand, Johannesburg 2050, South Africa
7 Interdisciplinary Centre for Water Research, Indian Institute of Science, Bangalore 560012,
India.
8 Planetary Science Institute, Tucson, AZ 85742 USA
9 Department of Earth, Energy, and Environment, University of Calgary, 2500 University
Drive NW
Calgary, AB, T2N 1N4





10  Department of Geology and Environmental Geosciences, University of Dayton, Dayton,
OH 45458, USA
Sustainability Program, University of Dayton, Dayton, OH 45458, USA
11 Department of Geography, National University of Singapore, Singapore

12 School of Chemical and Environmental Engineering, Technical University of Crete, Chania
73100, Greece

13 Rossby Centre, Swedish Meteorological and Hydrological Institute, Norrköping 60176,
Sweden

14 ADM Capital Foundation, Queen's Road Central, Hong Kong


Corresponding author: Stephan Harrison (Stephan.harrison@exeter.ac.uk) College of
Environment, Science and Economy, Exeter University, Exeter, UK





**Abstract**
In High Mountain Asia (HMA) ongoing climate change threatens mountain water resources
as glaciers melt, and the resulting changes in runoff and water availability are likely to have
considerable negative impacts on ecological and human systems. Numerous assessments of
the ways in which these glaciers will respond to climate warming have been published over
the past decade. Many of these assessments have used climate model projections to argue
that HMA glaciers will melt significantly this century. However, we show that this is only one
way in which these glaciers might respond. An alternative pathway is one in which
increasing valley-side instability releases large amounts of rock debris onto glacier surfaces.
The development of extensive glacier surface debris cover is common in HMA and, if thick
enough, this surface debris inhibits glacier melting to the extent that glacier ice becomes
preserved under the surface debris cover. In so doing, a transition to rock glaciers may
prolong the lifetime of HMA glaciers in the landscape. We call this alternative pathway the
Paraglacial Transition Model. In this Perspective Article we discuss the scientific basis of this
alternative view in order to better understand how HMA glaciers may respond to climate
change.
Key Words: High Mountain Asia. Glaciers. Paraglacial. Climate change.

**1 Introduction**
Understanding the current status, recent changes, and likely future evolution of glaciers in
High Mountain Asia (HMA) is important for a number of reasons. These include evaluating
the status of glacial water resources and how these may evolve under climate change
(Immerzeel et al. 2010; Rasul 2006; Lalande et al. 2021), and how glacier changes affect
glacial hazards (e.g. Harrison et al. 2018; Shugar et al. 2021). Given the likely warming by the
end of the twenty-first century in large parts of HMA, this understanding becomes a critical
issue as cryosphere-derived water supply affects the livelihoods of hundreds of millions of
people and the stability of ecosystems downstream. Eight of the 27 low-income and lower-
middle-income economies identified by the United Nations Development Programme in
Asia are currently affected by climate-driven water supply issues in the Himalaya and other
parts of HMA. The  Sustainable Development Goals (SDGs), adopted by all United Nations
(UN) and Asian governments, aim to substantially increase by 2030 the water-use efficiency
across all sectors, to ensure sustainable withdrawals and supply of freshwater, whilst also
reducing the number of people experiencing water scarcity (e.g.Bhaduri et al. 2016)). There
are also concerns about the impact of future glacier ice loss on global sea level change (e.g.
Marzeion et al. 2020) and on glacier-related hazards such as glacier lake outburst floods,
rock slides and falls and rapid changes in slope and catchment sediment yield (e.g. Li et al.
83    2022).



As a consequence of these concerns, there has been long-standing scientific and policy
focus on modelling changes in glacier mass balance and understanding the implications of
climate change for mountain water supplies (e.g.Nie et al. 2021). Numerous modelling
studies have projected the impacts of climate change on glacier mass loss in HMA (e.g.
Kraaijenbrink et al. 2017; Hock et al. 2019; Hugonnet et al. 2021; Rounce et al. 2020; 2023;
(Table 1) and downstream river runoff (e.g. Sorg et al. 2012; Lutz et al. 2014; Huss and Hock
2018). Since 2013, these have tended to use the outputs from CMIP5 set of model runs;
while the latest CMIP6 model runs are now available, few projections from this have so far
been employed.
Although existing modelling approaches are useful to assess pathways for future ice loss
from the region (Table 1), these generally assume that the different ways in which mountain
glaciers will evolve under future climate change has been accurately captured.  We argue
that this is not necessarily the case (Harrison et al. 2021). The common view is that the
expected rise in air temperature over this century is expected to lead to the almost
complete melting of glaciers in HMA by 2100 (e.g. Rounce et al. 2023; Chen et al. 2023),
thus that there is a simple deterministic and linear relationship between temperature
forcing and glacier mass balance response. Current modelling studies support substantial
but incomplete melting; for example, 60-98% reduction in glacier mass under RCP8.5 by the
end of the century according to Shannon et al. 2019. This arises from a combination of
reduction in accumulation as more precipitation falls in the form of rain and due to
enhanced melting associated with rising temperatures (see Table 1).

**Table 1**

There are, however, regional differences in mass loss projections across HMA and this partly
reflects model uncertainty at fine spatial scales (Chen et al. 2023)[0]. Kraaijenbrink et al.
(2017) used a global ensemble of 110 GCM runs from CMIP5 to assess the glacial response
driven by emissions under RCP2.6 scenarios and a consequent increase in Global Mean
Surface Temperature (GMST) of 1.5°C above pre-industrial conditions. This result suggests a
probable warming of 2.1±0.1°C for HMA by 2100, even at this low emissions scenario. They
also assessed likely regional changes and argued that parts of the western Pamir and the
Qilian Shan of northern China will lose most of its glacier mass compared with the present
day by 2100 with only 32±14% and 30±5% ice mass remaining, respectively. In this study,
the Karakoram region shows more resilience to climate warming, with a projected 80±7% of
ice volume remaining by 2100. This is attributed partly to the role of supraglacial debris
cover in maintaining ice mass, and the role of winter precipitation in maintaining
accumulation.



Supraglacial debris is now recognized as an important factor that may variably amplify or
buffer glacier mass balance response to temperature forcing (e.g. Herreid and Pellicciotti
2020; Shrestha et al. 2020; Chen et al. 2023; Pratap et al. 2023). Kraaijenbrink et al.(2017)
were among the first to model the impact of debris cover on glacier melt in HMA under
different climate projections and they showed that under RCP4.5, RCP6.0 and RCP8.5 glacier
mass losses would be 49±7%, 51±6% and 64±5%, respectively, by 2100 compared with the
present day. More recently, Compagno et al. (2022) used the five Shared Socioeconomic
Pathways (SSP119, SSP126, SSP245, SSP370, and SSP585) from CMIP6 to assess the future
evolution of debris cover and its impact on glacier dynamics for all HMA glaciers. They
showed a general increase in glacier debris cover with increasing radiative forcing, as well as
local increases in debris thickness on individual glaciers (see also Scherler et al. 2018; Molg
et al. 2020). At a smaller scale, Rowan et al. (2015) applied a numerical model to estimate
the evolution of the debris-covered Khumbu Glacier and predicted a decrease in glacier
volume of 8–10% by 2100.
Whilst such modelling experiments suggest high glacier volume loss in HMA by 2100, the
physical response of glaciers to climate change varies enormously across the Himalaya and
the wider HMA region and this is caused by varying exposure to monsoonal and westerly
atmospheric flows (e.g. Molg et al. 2014), changes in surface albedo and the variability of
local catchment characteristics such as local topography, aspect and geology (e.g. Fugger et
al. 2022).  For instance, glaciers experiencing accumulation in the summer months also
undergo ablation at this time (e.g. Fujita and Ageta 2000).  It is also known that the timing
and amount of monsoon snowfall can impact on mass balance in different regions of the
Himalaya, and for subsequent seasons (Molg et al. 2014; Bonekamp et al. 2019), and these
factors are not fully captured in existing climate models.
Despite this body of research we argue that more work needs to focus on likely geomorphic
responses to glacier mass loss across and within HMA if we are to better understand
landscape evolution during future deglaciation and any hydrological implications that
follow. The landscape responses that might increase in scale and spatial impact include rock
surface weathering; slope sediment supply and downslope sediment yield; mass
movements such as rock slope failures and debris flows; ecological succession and slope
greening; and their biophysical feedbacks (e.g. Knight 2024)  (Figure 1**). All such geomorphic
processes can deliver debris to glacier surfaces and surrounding slopes and valley floors,
thereby contributing to reduced melting through insulation of the ice beneath, alongside
downstream changes in sediment supply and changes in river transport capacity.



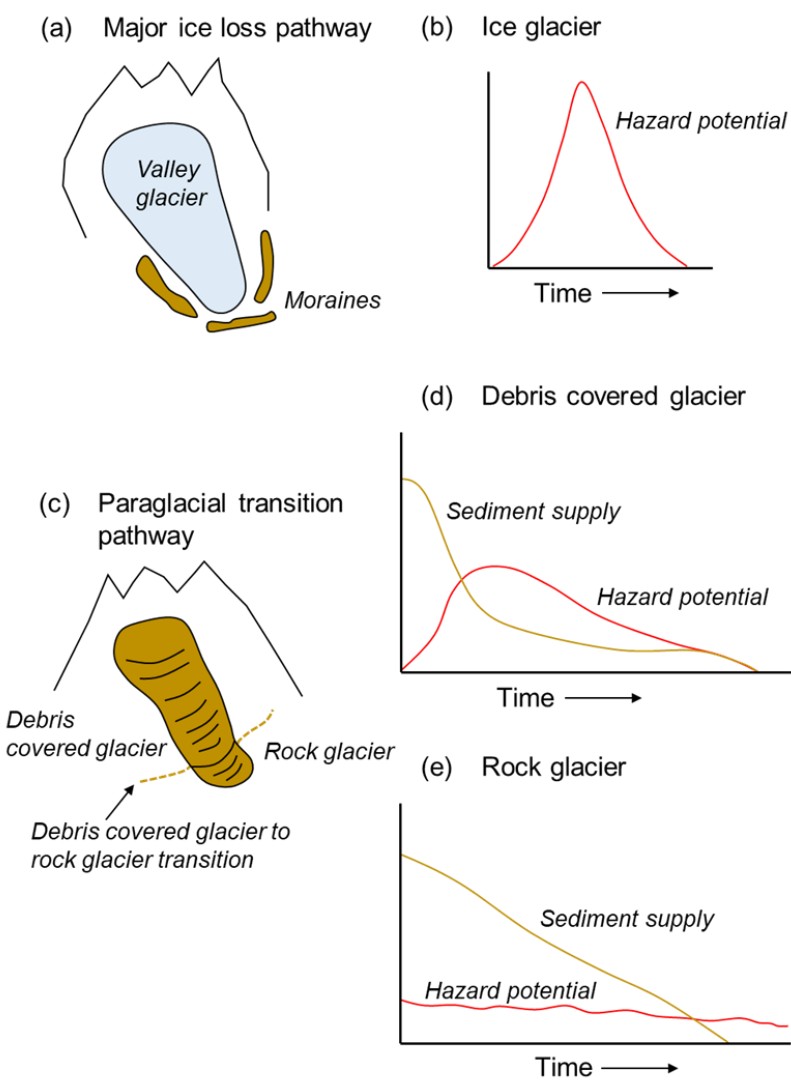



Figure 1. Schematic diagram showing the rapid increase and decrease in glacier hazards
during the Major ice loss pathway (a, b). Most of these hazards will be GLOFs. Hazards
associated with debris covered glaciers will show a rapid initial increase with a slowly falling
reduction (c, d). Hazards associated with rock glaciers will remain low and decrease over
time. The term 'Hazard potential' refers to the expectation that the probability of a named
hazard will change over time as the MIL or PT pathways evolve.




In this Perspective Article , we explore and highlight some of the implications of these
modelling exercises (the majority of which project sustained reduction of glacier mass
balance, e.g. Edwards et al. 2021) and highlight some plausible alternative scenarios for how
glacier systems in HMA might evolve to 2100 and beyond. We consider two broad scenarios
of how mountain glaciers might respond to climate change: the Major Ice Loss view (MIL)
and the Paraglacial Transition view (PT). Both of these end-member evolutionary pathways
necessarily represent simplifications of future glacier behaviour, yet they can usefully
explore how HMA glaciers could evolve over future decades. The two pathways highlight
the contingency of glacier evolution to the geomorphological, hypsometric, geological and
climatic variations that exist over HMA and that conventional climate model outcomes do
not successfully capture. It is already known, for instance, the glacier responses to recent
climate changes have been spatially and temporally varable across HMA (e.g. Rounce et al.
2020). Thus, our approach is grounded in an understanding of known glacier behaviour and
the properties of HMA glaciers, unlike the approach taken by climate models. Given these
caveats, we end by exploring some of these regional differences.
**2. Scenarios**
**2.1 Major Ice Loss (MIL) view**
As we have discussed, the conventional MIL view is that future climate warming will result
in widespread glacier recession and almost total ice loss in some parts of the Himalayas and
the wider HMA, particularly the eastern HMA (e.g.Shannon et al. 2019; Rounce et al. 2020;
2023). This is supported by the modelling projections made by the Glacier Model
Intercomparison Project (glacierMIP1(Hock et al. 2019) and the subsequent glacierMIP2.
The same understanding is reflected in the third phase (glacierMIP3), which is underway
and focuses on the equilibration of glaciers under various climatic conditions. GlacierMIP is
a coordinated intercomparison of global-scale glacier evolution models using standard initial
glacier conditions - glacier outlines from the RGI v6 inventory (Pfeffer et al. 2014; RGI 2017)
and ice thickness from Huss and Farinoti (2012) - forced with various GCMs under four
climate change scenarios. The participating glacier models varied in complexity: for
example, some models use temperature index schemes to calculate global-scale glacier
volume projections by 2100 while others use full energy balance models.  Models also differ
in the complexity with which glacier evolution is represented and each model therefore has
a bespoke approach to calibration that may impact on their comparability. The consensus
view from the glacierMIPs and other modelling studies, however, is that glaciers in the three
RGI regions covering HMA will experience significant reductions in ice volumes under the
business-as-usual RCP8.5 climate change scenario (Table 1). The potential trajectory of
evolution of HMA glaciers is shown diagrammatically in Figure 1.
**2.1.1 Impacts associated with the MIL view**



In essence, the MIL scenario eventually produces a HMA landscape consisting of much-
reduced glacier cover with small glaciers remaining at the highest altitudes and in some
niche locations. Associated with negative glacier mass balance and consequent glacier
retreat is the hypothesised increased frequency and magnitude of a number of glacier-
related hazards (e.g.Richardson and Reynolds 2000; Knight and Harrison 2014). Amongst the
most important of these at a local scale are GLOFs caused by the rapid drainage of glacial
lakes dammed by unstable moraines (e.g.Song et al. 2017; Nie et al. 2017; Emmer et al.
2022) and Landslide Lake Outburst Floods (LLOFs; Ruiz-Villanueva et al. 2017) caused by
breaching of lakes created by landslides. Other negative impacts at a regional scale include
ecosystem changes, warming of permafrost and subsequent rock mass collapse, the
potential reduction of water supplies downstream, increased seasonal discharge variability,
increased fluvial sediment fluxes and the impacts this has on agriculture, hydroelectric
power plants and dams in regional catchments (Immerzeel et al. 2010; Biemans et al. 2019;
Bosson et al. 2023).
Under this scenario, current glacier mass balance trends are exacerbated progressively over
time, leading to large numbers of proglacial lakes in overdeepened basins and dammed by
unstable moraines, and the slow melting of clean ice and debris-covered glaciers (e.g. Furian
2021). Locally these lakes are potentially hazardous, but by 2100 the HMA-wide GLOF peak
will have already been reached and will be subsiding (Harrison et al. 2018; Veh et al. 2019).
However, leading up to this end result would have been decades when GLOFs, LLOFs, large
debris flows and other mountain hazards became more frequent and, perhaps, larger than
in the recent historical period (e.g. Veh et al. 2020; Compagno et al. 2022).

**2.2 Paraglacial Transition (PT) view**

Despite the focus of much research on the MIL view, we argue that this misrepresents the
ways in which HMA glacier systems might evolve under climate warming because it largely
fails to reflect how glacial and mountain systems have responded in the past to deglaciation
from the last glacial maximum (e.g. Church and Ryder 1972; Ballantyne 2002; Cossart et al.
2007; Mercier et al. 2013; Knight and Harrison 2014), and are responding at present to
recent and ongoing climate warming (e.g. Knight et al., 2019).  This alternative view of the
future of HMA is that the glacial landscape will transition to a landscape dominated by
paraglacial processes, and we refer to this as the Paraglacial Transition (PT) view. Paraglacial
processes develop in response to deglaciation and are characterized, amongst others, by
increased rock slope failures from steep mountain slopes as these are de-buttressed by
glacier downwasting, and by increased debris flow activity from degrading lateral moraines
and related fluvial adjustment (see Li et al. 2022 for a review).
In the PT scenario, one end result is the potential for many stagnant clean ice glaciers to
become covered by rock debris and some of these undergoing renewed movement as their
termini evolve to form rock glaciers (Jones et al. 2019; Knight et al. 2019).




### 2.2.1 Impacts associated with the PT view

The PT scenario may eventually produce a HMA landscape dominated by relict ice masses of
different sizes and in different altitudinal and topographic settings, covered by varying
thicknesses of debris. This debris is released by enhanced weathering and rock slope failure
under a paraglacial process regime. The nature of the debris cover can give rise to a variety
of outcomes for buried ice masses. New rock glaciers can potentially develop as high-
mountain talus and other rock debris that presently is in extreme cold conditions enters a
condition where freeze-thaw is frequent and mobilized by periglacial processes. This would
represent a 'periglacial' path for the evolution of rock glaciers (e.g. Haeberli et al. 2006).  In
addition, we can see examples of such landscapes where debris-covered glaciers are
transitioning to ice-cored rock glaciers in other arid high mountains (Johnson 1980; Monnier
and Kinnard 2017).
Numerical modelling studies show the glacier debris-cover-rock glacier continuum (e.g.
Anderson et al. 2018), and we expect that rock glaciers in the Himalaya and other regions of
HMA will populate many of the currently glaciated valleys. However, none of the models
used in glacierMIP1 and the subsequent glacierMIP2 project consider a transition of ice or
debris covered glaciers into rock glaciers, and nor does IPCC AR6 (Hock et al. 2019). In
addition, while historically little has been written on these features in the Himalaya
(although much in other parts of HMA), recent research has shown that rock glaciers are
widely distributed in all parts of the Himalaya (Jones et al. 2021; Vishwakarma et al. 2022;
Harrison et al 2024) and that some ice glaciers and debris-covered glaciers are currently
undergoing a transition to form ice-cored rock glaciers (Jones et al. 2019).
If this PT scenario applies more widely then rock glaciers will eventually replace many clean-
ice and debris-covered glaciers as the main ice-bearing landforms in parts of the HMA,
perhaps alongside ice-cored moraines and ice-rich permafrost, with important implications
for future water supplies (e.g. Jones et al. 2021). This PT will likely increase the resilience of
ice bodies through increasing their longevity in the landscape.  Although research shows
that debris-covered glaciers are melting at similar rates to those without a substantial debris
cover (e.g. Pellicciotti et al., 2015), this appears to reflect high melt rates around
supraglacial ponds and declining ice discharge (e.g. Anderson et al. 2021).  Supraglacial
ponds are absent on RG, and the thick debris cover of these (Janke and Bolch 2021) will
extend the persistence of buried ice (see Figure 1).

This paraglacial path might result in a decreased GLOF hazard risk over time, although an
increased rock slope failure hazard. There may be some lakes impounded by rock glaciers
but these would expect to drain slowly rather than catastrophically given the armoured
nature of the rock dam.  Thus, the nature of geomorphic and hydrological hazards are
somewhat different between the MIL and PT pathways and this in itself may help



understand which glaciers are following which evolutionary pathway. GLOFs and LLOFs may
here also represent paraglacial landscape responses where, instead of meltwater passively
draining away under low hazard risk (MIL response), it is impounded by paraglacial
landslides that result in high hazard risk (PT response).

**3. Discussion and relationships between the MIL and PT pathways**

An important distinguishing feature between the MIL and PT pathways is that direct ice melt
is only a factor while the ice mass still exists, ceasing when the ice mass is gone, and tends
to affect local areas only. By contrast paraglaciation affects wider geographical areas
outside of the ice mass, and can extend for decades to millennia after full ice mass loss
(Ballantyne, 2002). Understanding which pathway of evolutionary development is followed
by any ice mass at any point in time has implications for predictability, hazard risk and
sustainable development. MIL and PT pathways represent end-members along an
evolutionary trajectory. It is likely that the development of any one glacier is dependent on
their initial conditions, and any changes in ice mass volume or surface debris (Figure 3).
However, both of these elements are more than a simple volumetric analysis because this
does not account for the detailed dynamics or spatial/temporal patterns of ice or debris.
These different pathways are likely to display important regional variation in the response
of clean ice glaciers, debris-covered glaciers and rock glaciers to future climate change in
HMA. These responses will not only be driven by variations in regional temperatures, but
also by changes in the behaviour of the Indian and East Asian summer monsoons, and the
westerlies (e.g. Fugger et al. 2022). Whilst regional climatic differences remain largely
unexplored (although see Brun et al. 2019), we can hypothesise that the areas that will
undergo a transition from ice glacier to rock glacier most readily will be those where debris-
covered glaciers are most common, because the debris supply in those areas is high.
Furthermore, as high-elevation clean ice thins, and as freeze-thaw and frost shattering
conditions move to higher altitudes, debris supply will increase in some high-elevation areas
that presently produce little debris. Where these places move into high-elevation cirques,
especially north-aspect cirques, 'periglacial' rock glacier development will be favoured (cf
Haeberli et al. 2006) Because of the exceptionally high relief of the Himalaya, Karakoram,
and many other parts of HMA, rock glacier development may proceed on regional scales not
seen in other mountains globally today, though it will take centuries for rock glaciers to
grow. Improved understanding of debris supply processes to valley bottoms is hampered by
the relative lack of modelling at regional scales that specifically considers the role of debris
cover on glacier dynamics, and mass balance (e.g. Racoviteanu et al. 2022).
How do we assess which of the MIL and PT scenarios are more likely and their probable
future spatial distributions? A first-order understanding might be gained by a simple
evaluation of how glaciers have behaved in the past in response to known climate forcings;
from this we can suggest whether these glaciers have shown high or low-sensitivity to past



or recent climate change (Harrison 2009).  Although this is a uniformitarian view, if the
glaciers have demonstrated a high sensitivity to past climate change then this tends to
support the MIL scenario for the future responses of glaciers to climate change.
Alternatively, if the glaciers have shown low sensitivity or delayed response to past climate
change then this might make the PT more likely, even if the forcings are different at
different times and glaciers today are in different states than they were in the past. To
explore this we need to establish the extent to which glaciers have responded to the
warming since the regional Last Glacial Maximum or the late Holocene Neoglacial Maximum
(often seen as equivalent to the European Little Ice Age).
Therefore, we compiled published studies that have dated Himalayan moraine sequences
from the Western, Central and Eastern Himalaya (Figure 2 and Supplementary Information
File). We analysed moraine ages from three time periods: the regional Last Glacial
Maximum from 18 to 24 ka (Owen et al. 2002), a period covering the regional Younger Dryas
from 12,880 and 11,640 ka (Rawat et al. 2012), and the regional Little Ice Age between
1300-1600 AD (Rowan 2017) (Figure 2). While there is evidence that glaciers in several areas
reached their late Pleistocene limits earlier than Marine Isotope Stage (MIS) 2 (Benn and
Owen 1998; Owen et al. 2002), overall these data show that glaciers in the Himalaya have
not receded far behind dated glacier limits over these time periods. Figure 3 shows that, as
expected, glaciers in different regions of the Himalaya have responded differently to past
climate change. The results are averaged by region and show considerable local variability.
However, overall, glaciers in the western and central regions of HMA have retreated less
over these time periods than those in the eastern part of the Himalaya; this might reflect
the reduction of the monsoon and intensification of westerly influences on glacier mass
balance towards the western Himalaya (Kumar et al. 2020; Hunt 2023).




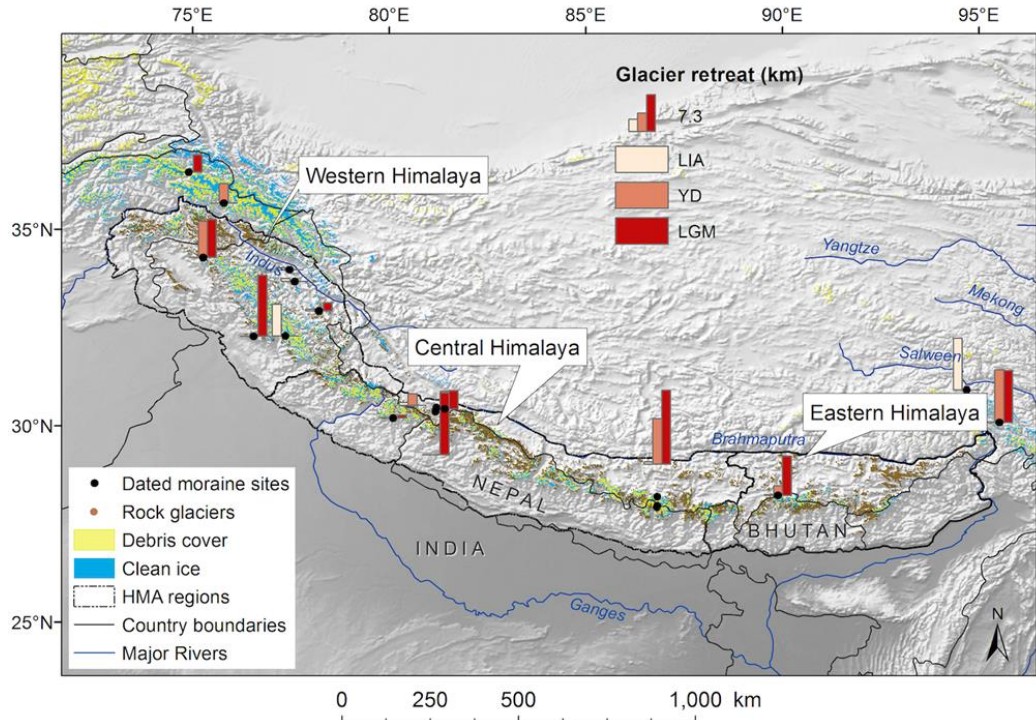

Figure 2. Distribution of the various types of glaciers (clean, debris-covered and rock
glaciers) across the HMA region. Clean ice outlines are based on the current RGI v6
inventory; debris-covered outlines are based on Herreid and Pelliciotti (2020) and rock
glacier locations (n = 24,968) (brown dots) from Jones et al. (2021). Also shown are the
dated moraine sites (black dots) complied for this study, and river systems (blue) (see
Supplementary Information file). Source: NextMap 2024.
One current limitation with this approach is that few moraines have been dated in HMA,
and most that have been are dated to the regional Neoglacial (Rowan 2017). Further, most
dated moraines are associated with fluctuations of large valley glaciers and therefore might
not reflect the behaviour of the more numerous and climatically sensitive smaller glaciers.
However, another way to assess glacier response to climate change in general (and
temperature rise in particular), and therefore the relative likelihood of the MIL or PT models
being dominant, is by monitoring their equilibrium line altitude (ELA), i.e. the altitude on the



glacier surface where the theoretical mass balance is zero at a given point in time (Zemp et
al. 2006; Cogley 2011). For instance, in the Khumbu region in the central Himalaya MIS 2
moraines (equivalent in age to the global Last Glacial Maximum) are located just 5 km or so
from modern ice limits and in many cases reflect a 200-300 m reduction in glacier ELA at this
time compared with current glacial ELAs in the region (Richards et al. 2000) and also
reduced insolation driving a weakened monsoon (Owen et al. 2002). At the western end of
the Himalaya much research has concentrated on the Nanga Parbat massif to the south of
the Karakoram. Here, work has shown that glaciers draining Nanga Parbat do not show an
MIS 2 maximum, although moraines of MIS 3 are present downvalley (Phillips et al. 2000).
The absence of evidence for an LGM-age advance of the glacier may reflect aridity during
MIS 2 in this region and therefore low glacier sensitivity to atmospheric temperatures at this
time (e.g. Yan et al. 2021), i.e. that glacier dynamics here is precipitation-controlled rather
than temperature-controlled.

Glacier recession since the regional Neoglacial maximum of the late 18[th] Century also
supports the present low sensitivity of many Himalayan glaciers to climate change (Rowan
2017). For instance, at Ama Dablam and Lhotse in the Khumbu region, present glacier
margins have retreated by around 1 km from Neoglacial moraine limits. Similarly, in the
monsoon-transition zone of the Indian Himalaya, the debris-covered Bara Shigri Glacier has
retreated less than 3 km since the 1850s (Chand et al 2017). Overall, Figure 3 demonstrates
that glacier termini in the western and central parts of the Himalaya have retreated less
than 2 km since the end of the Neoglacial maximum. Compared with considerable
volumetric ice loss since this time (Lee et al. 2021), if future glacier response mirrors that of
the past then this supports our contention that future glacier loss will dominantly involve
downwasting of glacier surfaces rather than terminus retreat. We argue that this favours
the development of stagnant glacier tongues and enhances the likelihood of further
transition to rock glaciers (Jones et al 2019) and thus the PT rather than the MIL pathway.

From this (albeit limited and incomplete) data set, we suggest that glaciers from the
western, eastern and northern Himalaya displayed low sensitivity to climate change during
the regional LGM and the Neoglacial, and this supports the PT scenario of glacier responses
to future climate change.

It is also likely that different glaciers in HMA are at different stages of deglacial evolution,
depending on their size, location and mass balance. This means that some show a MIL
response, some a PT response, and some may be changing from the former to the latter
(Figure 3). It is, however, uncertain as to which clean or debris-covered glaciers will
transition to rock glaciers and which will melt significantly in response to climate warming.
However, we can say, based on current observations, that small glaciers located above the
regional ELA are projected to disappear as they will not be able to adapt to future climate
(ICIMOD 2023), as is the case in the Andes (Ramirez et al. 2001) and the European Alps (e.g.



Zemp et al. 2006; Zebre et al. 2021). Some of these features may completely disappear if
debris supply is low, and others may undergo transitions to rock glaciers that may be
stabilized by a combination of high snowfall and high debris production. We hypothesise
that the transition process is dominated by debris fluxes from mountain slopes and the
connectivity between these sites and downwasting glacier surfaces below the ELA (Figure
3). Therefore, the transition between MIL and PT pathways is most efficient in areas where
high mountain slopes are producing rock slope failures, rock falls and debris flows, and
where lateral moraines are absent or poorly developed and therefore allow sediment access
from valley sides to the glacier surface. Climate change therefore will represent a first order
control on glacier behaviour but glacial processes creating lateral moraines will play a
significant second order control.
Finally, the consequences of having more persistent glacial ice in HMA would be profound.
This outcome would mean that there is more time to generate climate change mitigation
and adaptation schemes in the wider region, and develop technological fixes to the
challenges of changing hydrological resources (including total volumes and seasonality). In
addition, several of the UN Sustainable Development Goals such as Clean Water and
Sanitation might be more easily achieved than previously expected if cryospheric water
sources persist longer.

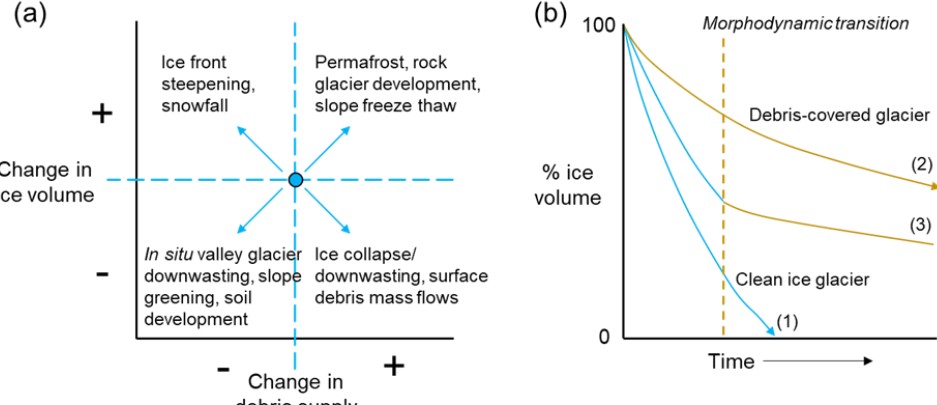

Figure 3. (a) Model of HMA glacier development as a consequence of certain changes in ice
volume vs changes in debris supply to/from the glacier surface. From an initial starting point
(blue circle), changes in ice volume and debris (blue arrows) are associated with certain
glacier properties and processes that suggest the likely ways in which these glaciers will
develop in future. The dashed blue lines represent zero balance in ice volume and debris



supply. (b) Representation of the different trajectories of changes in ice volume over time
between (1) a clean ice glacier (rapid melt), (2) a debris-covered glacier or rock glacier
(slower melt), and (3) a scenario where a clean ice glacier starts melting but is then covered
by debris, such as from a paraglacial landslide, that then slows down the ice melt. Such an
event represents a morphodynamic transition in the response of such a glacier to climate
forcing.
**3 Conclusions and future research imperatives**
Currently we have argued that there is a general consensus from climate modelling that
Himalayan and wider HMA glaciers will reduce their volume by up to 90% by 2100 in
response to projected warming, and most small glaciers will completely disappear by this
time. However, we have produced an alternative Paraglacial Transition scenario where
many glaciers transform into rock glaciers and other ice-debris landforms. This serves to
inhibit ice melt and increase the resilience of the Himalayan glacial system to future climate
change by increasing the longevity of ice bodies in the landscape. The Major Ice Loss (MIL)
and Paraglacial Transition (PT) scenarios discussed here represent end members of possible
glacial system responses to future climate change (Figure 3).
While the MIL scenario will also lead to a range of paraglacial responses from deglaciating
catchments, we argue that this will not necessarily change the future evolution of individual
glaciers which will continue to melt in response to ongoing climate warming (although
increased snowfall might reduce net mass loss). This MIL viewpoint continues to dominate
the literature based on climate model assessments of glacier melt. However, there are
relatively few published studies on the development of rock glaciers and their importance in
HMA (see Harrison et al. 2021 for a discussion of this). More research needs to be
conducted on the different ice masses and rock glaciers of HMA, and the paraglaciation of
the region if the PT view is to be properly assessed. Such future work is hampered by the
difficulty of assessing ice content in rock glaciers and other debris-covered landforms such
as lateral and terminal moraines, especially in remote, high-altitude settings. How many
rock glaciers have derived from the downwasting of glacial ice (Knight et al. 2019) and how
many are derived from the creep of ice-rich permafrost (e.g. Haeberli et al. 2024) is also
unknown. How rock glaciers respond to climate change in HMA is also hardly known given
their likely long response times.
Critical research is needed in order to evaluate the operations and outcomes of the MIL and
PT scenarios, and their possible interactions on individual glaciers. Future research
imperatives therefore include: 1) determination of debris fluxes throughout the region for
the full range of geological materials, slopes, and microclimates and glacier types; 2) long-
term monitoring of glacier mass balance across the region in order to evaluate cryospheric
sensitivity to climate forcing; 3) measurement of contemporary debris fluxes and
distributions on different glacier types; 4) present and past climate modelling with snow
accumulation with concurrent debris loading, and 5) projections into the future for the full




range of climate scenarios. Development of ultra-downscaled climate modelling that is
responsive to the full range of HMA relief and slopes with resolutions enough to resolve
individual cirques is also needed. This may be currently possible for small local geographic
domains sufficient to sample different parts of HMA.


| | | Marzeion et al., 2012* | Giesen and Oerlemans, 2013* | Hirabayashi et al., 2013* | Radić et al., 2014* | Huss and Hock, 2015* | Shannon et al., 2019** | Rounce et al. 2023* |
|---|---|---|---|---|---|---|---|---|
| Central Asia | | 63.7±6.8 | 67.2±8.7 | 61.0±6.6 | 73.6±11.0 | 88.3±7.8 | 80.0±7.0 | 80.0± 17.0% |
| South Asia West | | 43.1±6.2 | 78.1±10.4 | 57.5±5.6 | 62.7±15.2 | 84.0±13.7 | 98.0±1.0 | 69.0±20. 0% |
| South Asia East | | 62.9±8.2 | 93.7±4.3 | 42.3±8.5 | 76.4±9.9 | 86.0±24.2 | 95.0±2.0 | 94.0±4.0 % |


**Table 1** Examples of projected relative mass losses by the end of 21st Century for HMA,
from different recent studies (reduction as a percentage of ice loss from 1990). Regions are
defined as in Randolph Glacier inventory (RGI) v6 (first-order region, shown in Figure 2). The
values refer to the multi-GCM means and their standard deviation.

*\* denotes the projections generated by glacierMIP1 using CMIP5 RCP8.5 climate forcing.*
*\*\* denotes projections made with downscaled CMIP5 RCP8.5 model for high-end climate*
*scenarios.*

Code/Data availability
Data are available in the Supplementary Information File (moraine inventory) and at the zenodo
link(10.5281/zenodo.11237094) (rock glacier inventory).




Author contribution
SH developed the initial idea and wrote the first draft of the paper.  The paper was developed with
the insights of AR, NFG, KA, JK, UH, DS and  JK .  DJ produced the rock glacier inventory and ARanger
produced the moraine inventory.  JK developed figures 1 and 3 and AR developed figure 2.  All
authors contributed to the development of the paper.
Competing interests
We have no competing interests.



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
