# Peer review of "Will landscape responses reduce glacier sensitivity to climate change in High Mountain Asia? 2 Stephan Harrison1, Adina Racoviteanu2, Sarah Shannon3, Darren Jones1, Karen Anderson4, 3 Neil Glasser5, Jasper Knight6, Anna Ranger1, Arindan Mandal7, Brahma Dutt Vishwakarma7, 4 Jeffrey S. Kargel8, Dan Shugar9, Umesh Haritashya10, Dongfeng Li11, Aristeidis Koutroulis12, 5 Klaus Wyser13, Sam Inglis14 6 7"

_EGUsphere, 2024_

## Referee Comment (RC1)

**Review of 'Will landscape responses reduce glacier sensitivity to climate change in High Mountain Asia?'**

**General comments**

The manuscript presents a detailed consideration of an alternative to the assumed evolution of climatic change in High Mountain Asia that is the focus of numerical modelling to date. The manuscript focuses on a more geomorphological approach of glacial landscape change in the coming century in the region, and contributes an important piece of work in an important area of glacial research.

The manuscript it written is a clear and accessible writing style. I have suggested some rewording of sentences to increase the applicability to a global audience and the general flow of the article. I think the figures, figure captions and references to figures would benefit from some editing, which I provide details of in the 'Specific Comments' section.

In conclusion, I think this manuscript is an important contribution to research in the area of glacial change in HMA, and is starting a key area of discussion that needs to be had in a field where regional scale numerical modelling has dominated over the past decade. I think this manuscript is publishable following minor to moderate corrections, detailed below.

**Specific comments**

1. The three figures included in the manuscript all contribute to the manuscript, but I think their placement does not support the text as fully as they could. Furthermore, the references to the figures in the text is brief and it is often not very clear how the figure supports the statement made. Suggestions to improve these points below:
    a. For Figure 1, I suggest removing (a) and (c) – I do not feel these contribute to the figure, and I think they detract from the more important information in (b), (d) and (e).
    b. When referred to in the text, the descriptions of figures about what needs to be taken from the figure is often vague. I suggest adding further words to explain fully what the reader needs to look for that helps them in understanding the argument made. It was difficult to confirm whether you were referring to the correct figures in a number of cases as it was unclear what was needed to be drawn from the figure.

2. Throughout the manuscript there are references to varying terminology for regions within HMA. I think it would be worth reviewing this terminology and either reducing the number of regions referred to or including them all in Figure 2 so the reader is aware of where you are referring to. I think it would also be worth placing, and referring to, this figure earlier in the manuscript so readers are aware of the regions you are referring to.

3. For Figure 3b, I think it would be worth considering whether this graph would be better placed in Figure 1, so all the graphs presented could be compared more easily. I also suggest labelling the (3) line on the graph along the lines of 'Clean-debris transition glacier', so all elements of the graph are clearly understood.

4. Please look at the placement of figures and the references to these figures throughout the text. Figure 3 was referred to before Figure 2.

5. A great effort has been made throughout to write short, concise sentences. However, there is frequent use of 'This' is the following sentence when this approach is taken. I suggest rereading the manuscript to make sure it is clear what 'this' is referring to and amending where necessary to make the focus of some sentences clearer – in some cases I have suggested additional words or rewording within the technical corrections.

6. The paragraph from Lines 135 to 144 does not fit where it is place currently, and the information within it would be better suited earlier on in the section so when 'winter precipitation' is referred to on Line 119 this makes sense. I suggest removing the first sentence of this paragraph and placing the paragraph starting on Line 108 instead.

The final sentence (Lines 497-498) is key here, as it will help future researchers to show the rationale for geomorphological based fieldwork. I suggest some rewording by replacing 'This' with 'Such an approach' but also including a 'but' at the end of the sentence – something that makes a nod towards the difficultly of such data collection being undertaken.

**Technical corrections**

Line 67: Suggest including some examples of the 'number of reasons' referred to here.

Line 70: Include 'related' following 'glacial'.

Line 72: Suggest quantifying the 'hundreds of millions of people' referred to here to show the importance of the research.

Line 75: Suggest removal of 'Himalaya and other parts' to make the sentence more concise.

Line 77: Suggest moving 'by 2030' to the end of the sentence to make flow of sentence better.

Line 79: Remove extra bracket after citation.

Line 81 and 82: Suggest inclusion of citation after 'glacier lake outburst floods' and ' rock slides and falls' to provide evidence as you have for the other types of change mentioned in the sentence.

Line 87: A space is needed between 'e.g.' and 'Nie'.

Line 90: Suggest inclusion of 'the' before 'CMIP5.

Line 99: I think a word such as 'direct' or 'simple' rather than 'linear', as I do not think this is always the case.

Line 102: Place '2019' in brackets.

Line 102: Suggest inclusion of 'outcome' between 'This' and ' arises'.

Line 108: Suggest moving 'however' to the start of the sentence.

Line 108: Suggest replacing 'and this' with 'which.

Line 109: Remove superscripted 0 after citation.

Line 115: Suggest replacing 'with' with 'to'.

Line 118: Include 'anomaly' between 'This' and 'is'.

Line 123: Replace 'et_al.(2017)' with 'et al. (2017)'.

Line 145: Suggest including a comma after 'research'.

Lines 149-151: Suggest remove of semi colons in this sentence and replacing with a comma.

Line 151: Make the closing bracket normal text – it is currently in bold.

Line 159: GLOF needs to be referred to in full here, as Glacial Lake Outburst Floods have not been referred to before this point.

Line 158-163: Please include reference to (e) in the figure caption.

Line 165: Remove space before comma.

Lines 166-167: Suggest removing the opening bracket with a comma  on this line and placing the opening bracket before 'e.g.' instead.

Line 175: Suggest inclusion of 'that' after 'instance,'.

Line 178: Suggest inclusion of 'more generalised' before 'approach'.

Line 182: Suggest removal of 'As we discussed,'.

Lines 183-184: Suggest the removal of 'the Himalaya and the wider' here to make references to the region in question clearer.

Line 185: Suggest replacement of 'This is' with 'The conclusions are'.

Line 186: Remove second opening bracket.

Line 187: Check 'glacierMIP3' does not need a citation.

Line 190: Define 'RGI' as this is the first time it is referred to.

Line 191:  Suggest replacing hyphen with a comma.

Lines 192-193: Suggest replacing ': for example,' with 'with.

Line 196: Suggest including 'However,' before 'the consensus'.

Line 198: Suggest naming the three HMA RGI regions here to link clearly with the figure.

Lines 199-200: The figure caption for Figure 1 does not explicitly say it is for HMA glaciers. I suggest either rewording this sentence (maybe including ' which applies to' or similar) or including HMA in the figure caption. It is then clear whether Figure 1 is specific to HMA glaciers or whether it applies to glacier globally and is being using to make a point about HMA glaciers here.

Line 206: Add space between 'e.g.' and 'Richardson'.

Line 207: Suggest defining GLOF here, as it is the first use of it outside a figure caption.

Line 208: Add space between 'e.g.' and 'Song'.

Line 220: Suggest changing 'will have been' to 'is to projected to have been'.

Line 221: Suggest replacing 'leading up to this result' with 'up to this point, there'.

Line 222: Suggest replacing 'became' with 'were'.

Line 222; Suggest replacing ',perhaps,' with 'potentially'.

Line 225: Suggest including ' approach' before 'misrepresents'.

Line 227: Suggest adding 'geomorphologically' after 'responded'.

Line 228: Capitalise Last Glacial Maximum.

Line 233: Suggest removing  ',amongst others' and include 'processes such as' between 'by' and 'increased'.

Line 235: Suggest removal of 'by'.

Line 238: Suggest replacement of 'and' with ',with'.

Line 244: Suggest inclusion of 'rock' before debris.

Line 244: Suggest replacing 'This' with 'Rock'.

Line 247: Suggest replacing 'that presently is' with 'which, at present;.

Lines 247-248: Suggest removing 'extreme cold conditions' and 's' from 'enters' and adding 'due to extreme conditions at the end of the sentence.

Line 248: Suggest replacing 'This' with 'If these conditions persist'.

Line 253: Suggest replacing 'the' with 'a'.

Line 255: Suggest adding 'in the PT scenario' at the end of the sentence.

Line 258: Suggest moving 'historically' to be after 'Himalaya'.

Line 259: Suggest replacing 'much' with more so' and including examples of this point in the form of citations.

Line 263: Suggest including a comma after 'widely'.

Line 265: Suggest replacing 'perhaps' with 'potentially'.

Line 269: Suggest inclusion of 'parity' or similar word between 'this' and 'appears'.

Line 271: Replace 'RG' with 'RGI'.

Line 275: Suggest replacing 'This paraglacial pathway' with ' The PT scenario/pathway'.

Line 276: Suggest including 'is anticipated' at the end of the sentence.

Lines 277-278: Suggest more explanation of the statement 'given the armoured nature of the rock dam' in this context.

Lines 278-280: Suggest this sentence goes last in the paragraph as it makes an important conclusionary point.

Line 295: Remove comma,

Line 294: This is the first reference of Figure 3, so I would expect the figure to be place at the end of the paragraph. Also, Figure 2 has not yet been referred to.

Line 301: Check Westerlies are referred to in the overview of meteorological/climate conditions in the HMA, so the reference to it here make sense.

Line 302: Suggest replacing 'although see' with 'with the exception of'.

Line 304: Suggest including 'already' before 'most.

Line 305: Suggest inclusion of 'conditions that enable' before freeze-thaw.

Line 306: Suggest replacing of 'conditions' with 'to occur'.

Line 307: Suggest replacing 'places' with 'conditions'.

Line 308: Include Karakoram on Figure 2.

Line 310: Suggest replacing 'not seen' with 'greater than those seen'.

Line 311: Suggest including 'these' before 'rock glaciers'.

Line 312: Suggest replacing 'grow' with 'develop'.

Line 312: Suggest replacing 'hampered' with 'restricted'.

Lines 315-316: Suggest placing 'A first order understanding of which od the MIL and PT scenarios are more likely and their future spatial distributions' at the start of the sentence and removing the sentence question completely.

Line 320: Suggest including 'in question' after 'glaciers'.

Lines 320-321 and Lines 323: For the statements made about the MIL and PT scenarios here, I think further explanation of these statements would be of benefit here.

Line 328: Suggest replacing 'Therefore,' with 'To achieve this' or 'Consequently'.

Line 336: Suggest replacing 'far behind' with 'much further than'.

Line 336: Suggest replacing 'over' with 'from'.

Line 340; Suggest replacing 'these' with 'the investigated'.

Line 340: Suggest replacing ';this' with '. This spatial disparity'.

Line 365: Suggest replacing ';' with ', whilst'.

Lines 366 and 367: Suggest combing the two bracketed pieces of information in both cases for clearer reading.

Line 368: Suggest replacing 'Source:' with 'Adapted from'.

Line 372: Suggest replacing 'few' with 'that', following 'moraines'.

Line 372: Suggest including 'are limited,' after HMA.

Line 373: Include comma between 'Rowan' and '2017'.

Line 380: Capitalise Equilibrium Line Altitude.

Line 382: Include comma after Himalaya.

Line 386: It would be beneficial to include explanation of the statement of about the weakened monsoon and include its effect on accumulation explicitly.

Line 391: Suggest replacing 'this' with 'the'.

Lines 397-298: Suggest moving 'at Ama Dablam and Lhotse in the Khumbu region' to the end of the sentence and replacing 'at' with 'for'.

Line 403: Include comma in citation.

Line 403: Suggest including 'predicted' before 'future' and replacing ' will dominantly involve' with 'must have been dominated by'.

Line 409: Suggest replacing 'this' with 'the'.

Line 409: Suggest including 'presented here, despite being limited and incomplete,' after 'data set' and removing '(albeit limited and incomplete)'.

Line 411: Suggest replacing 'and this' with 'which'.

Lines 409-412; Suggest combing the paragraph with the following paragraph.

Line 420: Suggest replacing ' regional ELA' with 'regionally averaged or 'more regional'.

Line 421: Include commas in citations.

Line 422: Include commas in citations.

Line 430: Suggest swapping 'therefore will' around.

Line 454: Suggest replacing 'that' with 'which'.

Line 460: Suggest replacing 'climate' with 'numerical' and including 'results' after modelling.

Line 462: Suggest including 'climate' before 'warming'.

Line 463: Suggest replacing 'produced' with 'presented'.

Line 464: Suggest including 'as climate change progresses' at the end of the sentence.

Line 470: Suggest removing 'also'.

Line 473: Suggest replacing 'viewpoint' with 'scenario' or 'suggested scenario'.

Line 478: Suggest replacing 'hampered' with 'made challenging' or similar.

Line 483: Suggest replacing 'hardly known' with 'poorly understood',

Line 483: Suggest including ', particularly' before 'given'.

Line 493: Suggest replacing 'on' with 'for'.

Line 496: Suggest replacing 'enough' with 'sufficiently detailed' and placing it before 'resolutions'.

---

## Author Comment (AC1)

**Will landscape responses reduce glacier sensitivity to climate change in High Mountain Asia?**

Response to Reviewer 1

Reviewer 1

We thank the reviewer 1 for their detailed comments and suggestions on our paper. These have been extremely useful in helping us rewrite the text and have made the paper much clearer. Our replies are in red below.

**Specific comments**
The three figures included in the manuscript all contribute to the manuscript, but I think their placement does not support the text as fully as they could. Furthermore, the references to the figures in the text is brief and it is often not very clear how the figure supports the statement made. Suggestions to improve these points below: For Figure 1, I suggest removing (a) and (c) – I do not feel these contribute to the figure, and I think they detract from the more important information in (b), (d) and (e).
When referred to in the text, the descriptions of figures about what needs to be taken from the figure is often vague. I suggest adding further words to explain fully what the reader needs to look for that helps them in understanding the argument made. It was difficult to confirm whether you were referring to the correct figures in a number of cases as it was unclear what was needed to be drawn from the figure.
We have reviewed this suggestion and have made the Figure captions more informative. However, having discussed the Figure we do feel that the Figure 1 a-c provides a graphical description of how we expect some glaciers to behave and this clearly illustrates this. The alternative would be to provide some exemplar photographs or other images but we feel that this would not work as well.

1. Throughout the manuscript there are references to varying terminology for regions within HMA. I think it would be worth reviewing this terminology and either reducing the number of regions referred to or including them all in Figure 2 so the reader is aware of where you are referring to. I think it would also be worth placing, and referring to, this figure earlier in the manuscript so readers are aware of the regions you are referring to.
   We have done this by making sure that we refer to HMA more consistently throughout the text.

1. For Figure 3b, I think it would be worth considering whether this graph would be better placed in Figure 1, so all the graphs presented could be compared more easily. I also suggest labelling the (3) line on the graph along the lines of 'Clean-debris transition glacier', so all elements of the graph are clearly understood.

1. Please look at the placement of figures and the references to these figures throughout the text. Figure 3 was referred to before Figure 2.
   We have now addressed this issue.

1. A great effort has been made throughout to write short, concise sentences. However, there is frequent use of 'This' is the following sentence when this approach is taken. I suggest rereading the manuscript to make sure it is clear what 'this' is referring to and amending where necessary to make the focus of some sentences clearer – in some cases I have suggested additional words or rewording within the technical corrections. Thank you for pointing this out. As you see, we have accepted many of your suggestions.

1. The paragraph from Lines 135 to 144 does not fit where it is place currently, and the information within it would be better suited earlier on in the section so when 'winter precipitation' is referred to on Line 119 this makes sense. I suggest removing the first sentence of this paragraph and placing the paragraph starting on Line 108 instead.

The final sentence (Lines 497-498) is key here, as it will help future researchers to show the rationale for geomorphological based fieldwork. I suggest some rewording by replacing 'This' with 'Such an approach' but also including a 'but' at the end of the sentence – something that makes a nod towards the difficultly of such data collection being undertaken. We have made the changes to the structure discussing climate and modelling as you suggest.

**Technical corrections**

Line 67: Suggest including some examples of the 'number of reasons' referred to here. Done

Line 70: Include 'related' following 'glacial'. Done

Line 72: Suggest quantifying the 'hundreds of millions of people' referred to here to show the importance of the research. Difficult to do this as different authors identify different numbers. That's why we kept this rather vague.

Line 75: Suggest removal of 'Himalaya and other parts' to make the sentence more concise. Done

Line 77: Suggest moving 'by 2030' to the end of the sentence to make flow of sentence better. Done

Line 79: Remove extra bracket after citation. Done

Line 81 and 82: Suggest inclusion of citation after 'glacier lake outburst floods' and ' rock slides and falls' to provide evidence as you have for the other types of change mentioned in the sentence. These are covered in the Li et al reference.

Line 87: A space is needed between 'e.g.' and 'Nie'. Done

Line 90: Suggest inclusion of 'the' before 'CMIP5. Done

Line 99: I think a word such as 'direct' or 'simple' rather than 'linear', as I do not think this is always the case. Done

Line 102: Place '2019' in brackets. Done

Line 102: Suggest inclusion of 'outcome' between 'This' and ' arises'. Done

Line 108: Suggest moving 'however' to the start of the sentence. Done

Line 108: Suggest replacing 'and this' with 'which. Done

Line 109: Remove superscripted 0 after citation. Done

Line 115: Suggest replacing 'with' with 'to'. Done

Line 118: Include 'anomaly' between 'This' and 'is'. Done

Line 123: Replace 'et_al.(2017)' with 'et al. (2017)'. Done

Line 145: Suggest including a comma after 'research'. Done

Lines 149-151: Suggest remove of semi colons in this sentence and replacing with a comma. We prefer semi-colons when the list is long.

Line 151: Make the closing bracket normal text – it is currently in bold. Done

Line 159: GLOF needs to be referred to in full here, as Glacial Lake Outburst Floods have not been referred to before this point. Done

Line 158-163: Please include reference to (e) in the figure caption. Done

Line 165: Remove space before comma. Done

Lines 166-167: Suggest removing the opening bracket with a comma on this line and placing the opening bracket before 'e.g.' instead. Done

Line 175: Suggest inclusion of 'that' after 'instance,'. Done

Line 178: Suggest inclusion of 'more generalised' before 'approach'. Done

Line 182: Suggest removal of 'As we discussed,'. Done

Lines 183-184: Suggest the removal of 'the Himalaya and the wider' here to make references to the region in question clearer. Done

Line 185: Suggest replacement of 'This is' with 'The conclusions are'. Done

Line 186: Remove second opening bracket. Done

Line 187: Check 'glacierMIP3' does not need a citation. Done

Line 190: Define 'RGI' as this is the first time it is referred to. I don't think this is needed

Line 191: Suggest replacing hyphen with a comma. Done

Line 196: Suggest including 'However,' before 'the consensus'.Done

Line 198: Suggest naming the three HMA RGI regions here to link clearly with the figure. Done

Lines 199-200: The figure caption for Figure 1 does not explicitly say it is for HMA glaciers. I suggest either rewording this sentence (maybe including ' which applies to' or similar) or including HMA in the figure caption. It is then clear whether Figure 1 is specific to HMA glaciers or whether it applies to glacier globally and is being using to make a point about HMA glaciers here. Done

Line 206: Add space between 'e.g.' and 'Richardson'. Done

Line 207: Suggest defining GLOF here, as it is the first use of it outside a figure caption. Done

Line 208: Add space between 'e.g.' and 'Song'. Done

Line 220: Suggest changing 'will have been' to 'is to projected to have been'. Disagree.

Line 221: Suggest replacing 'leading up to this result' with 'up to this point, there'. Done

Line 222: Suggest replacing 'became' with 'were'. We think that this is OK.

Line 222; Suggest replacing ',perhaps,' with 'potentially'. Done

Line 225: Suggest including ' approach' before 'misrepresents'. Done

Line 227: Suggest adding 'geomorphologically' after 'responded'. Done

Line 228: Capitalise Last Glacial Maximum. Done

Line 233: Suggest removing ',amongst others' and include 'processes such as' between 'by' and 'increased'. Done

Line 235: Suggest removal of 'by'. Done

Line 238: Suggest replacement of 'and' with ',with'. Disagree

Line 244: Suggest inclusion of 'rock' before debris. Done

Line 244: Suggest replacing 'This' with 'Rock'. Disagree because it also is made up of fine sediment. Text changed to reflect this.

Line 247: Suggest replacing 'that presently is' with 'which, at present;. Done

Lines 247-248: Suggest removing 'extreme cold conditions' and 's' from 'enters' and adding 'due to extreme conditions at the end of the sentence. Disagree

Line 248: Suggest replacing 'This' with 'If these conditions persist'. Done

Line 253: Suggest replacing 'the' with 'a'. Disagree, it might be several

Line 255: Suggest adding 'in the PT scenario' at the end of the sentence. Disagree

Line 258: Suggest moving 'historically' to be after 'Himalaya'. Disagree

Line 259: Suggest replacing 'much' with more so' and including examples of this point in the form of citations. Done

Line 265: Suggest replacing 'perhaps' with 'potentially'.Done

Line 269: Suggest inclusion of 'parity' or similar word between 'this' and 'appears'. Disagree

Line 271: Replace 'RG' with 'RGI'. Done

Line 275: Suggest replacing 'This paraglacial pathway' with ' The PT scenario/pathway'. Done

Line 276: Suggest including 'is anticipated' at the end of the sentence. Disagree

Lines 277-278: Suggest more explanation of the statement 'given the armoured nature of the rock dam' in this context. Done

Lines 278-280: Suggest this sentence goes last in the paragraph as it makes an important conclusionary point. Done

Line 295: Remove comma, Done

Line 294: This is the first reference of Figure 3, so I would expect the figure to be place at the end of the paragraph. Also, Figure 2 has not yet been referred to. Done

Line 301: Check Westerlies are referred to in the overview of meteorological/climate conditions in the HMA, so the reference to it here make sense. Done

Line 302: Suggest replacing 'although see' with 'with the exception of'. Disagree

Line 304: Suggest including 'already' before 'most. Done

Line 305: Suggest inclusion of 'conditions that enable' before freeze-thaw. Disagree

Line 306: Suggest replacing of 'conditions' with 'to occur'. Done

Line 307: Suggest replacing 'places' with 'conditions'. Done

Line 308: Include Karakoram on Figure 2. Deleted

Line 310: Suggest replacing 'not seen' with 'greater than those seen'. Done

Line 311: Suggest including 'these' before 'rock glaciers'. Disagree

Line 312: Suggest replacing 'grow' with 'develop'. Done

Line 312: Suggest replacing 'hampered' with 'restricted'. Done

Lines 315-316: Suggest placing 'A first order understanding of which od the MIL and PT scenarios are more likely and their future spatial distributions' at the start of the sentence and removing the sentence question completely. Disagee, this is an opinion piece

Line 320: Suggest including 'in question' after 'glaciers'. Disagree

Lines 320-321 and Lines 323: For the statements made about the MIL and PT scenarios here, I think further explanation of these statements would be of benefit here. Disagree. We are not sure what is suggested.

Line 328: Suggest replacing 'Therefore,' with 'To achieve this' or 'Consequently'. Done

Line 336: Suggest replacing 'far behind' with 'much further than'. Done

Line 336: Suggest replacing 'over' with 'from'. Disagree

Line 340; Suggest replacing 'these' with 'the investigated'. Disagree

Line 340: Suggest replacing ';this' with '. This spatial disparity'. Done

Line 365: Suggest replacing ';' with ', whilst'. Disagree

Lines 366 and 367: Suggest combing the two bracketed pieces of information in both cases for clearer reading. Done

Line 368: Suggest replacing 'Source:' with 'Adapted from'. Done

Line 372: Suggest replacing 'few' with 'that', following 'moraines'. Disagree

Line 372: Suggest including 'are limited,' after HMA. Disagree

Line 373: Include comma between 'Rowan' and '2017'. Done

Line 380: Capitalise Equilibrium Line Altitude. Done

Line 382: Include comma after Himalaya. Done

Line 386: It would be beneficial to include explanation of the statement of about the weakened monsoon and include its effect on accumulation explicitly. Disagree

Line 391: Suggest replacing 'this' with 'the'. Done

Lines 397-298: Suggest moving 'at Ama Dablam and Lhotse in the Khumbu region' to the end of the sentence and replacing 'at' with 'for'. Disagree

Line 403: Include comma in citation. Done

Line 403: Suggest including 'predicted' before 'future' and replacing ' will dominantly involve' with 'must have been dominated by'. Disagree

Line 409: Suggest replacing 'this' with 'the'. Done

Line 409: Suggest including 'presented here, despite being limited and incomplete,' after 'data set' and removing '(albeit limited and incomplete)'. Done

Line 411: Suggest replacing 'and this' with 'which'. Done

Lines 409-412; Suggest combing the paragraph with the following paragraph. Done

Line 420: Suggest replacing ' regional ELA' with 'regionally averaged or 'more regional'. Disagree

Line 421: Include commas in citations. Done

Line 422: Include commas in citations. Done

Line 430: Suggest swapping 'therefore will' around. Done

Line 454: Suggest replacing 'that' with 'which'. Disagree

Line 462: Suggest including 'climate' before 'warming'. Done

Line 463: Suggest replacing 'produced' with 'presented'. Done

Line 464: Suggest including 'as climate change progresses' at the end of the sentence. Done

Line 470: Suggest removing 'also'. Disagree

Line 473: Suggest replacing 'viewpoint' with 'scenario' or 'suggested scenario'. Done

Line 478: Suggest replacing 'hampered' with 'made challenging' or similar. Done

Line 483: Suggest replacing 'hardly known' with 'poorly understood', Disagree

Line 483: Suggest including ', particularly' before 'given'. Done

Line 493: Suggest replacing 'on' with 'for'. Disagree

Line 496: Suggest replacing 'enough' with 'sufficiently detailed' and placing it before 'resolutions'. Disagree

---

## Author Comment (AC2)

**Will landscape responses reduce glacier sensitivity to climate change in High Mountain Asia?**

We thank the reviewer 2 for their detailed comments and suggestions on our paper. These have been extremely useful in helping us rewrite the text and have made the paper much clearer. Our replies are in red below.

Response to Reviewer 2

**Summary**

The authors present a perspective on an alternative pathway (called the 'Paraglacial Transition' view) of how Himalayan glacier systems may respond to climate change in the future, different to the common view that future climate warming will result in sustained glacier retreat and ice loss ('Major ice loss' view). They describe how rising rock and moraine instabilities release inscreasing amounts of debris material on glacier surfaces, with impacts on their melt, dynamics, and morphologies. A preliminary analysis of past long-term glacier retreat rates shows distinct differences across HMA. The authors suggest a potential transition into stagnant debris covered glacier tongues and eventually into rock glaciers for some glaciers, causing a prolonged life cycle for glacier ice.

**General comments**

The manuscript is well written and structured, and the balance of theoretical background, motivation and new insights based on a preliminary analysis is tempting and adequate. The study gives a fresh view on alternative ways how glaciers in the Himalays can develop in the future, complementary to the conventional view based on global and regional-scale simulations of sustained ice loss in the Himalayas expected by the end of the century. I like the authors' view on future glacier response as mirroring that of the past. In my opinion the manuscript fits well into the scope of The Cryosphere and as a perspective paper. However, I suggest addressig two major points and clarify few minor points specified below, to make the study more robust.

We thank the reviewer for these supportive words.

**Main points**

- Debris thickness

I miss a discussion of the fact that the major part of debris supply for debris-covered tongues comes from headwalls and is transported englacially, i.e. its rate is controled by headwall erosion rates, but also the glacier's ice dynamics and surface ablation. Therefore, I don't see clearly how e.g. typically downwasted (concave) near-stagnant debris-covered tongues can become drastically more debris-covered and eventually transition into a rock glacier in the near or far future. Unless the authors are talking in general about very small

tongues of debris-covered glaciers without lateral moraines, which might be directly connected gravitationally to nearby rock walls. But in general, I think lateral debris supply from headwall erosion, rock falls and moraines can not efficiently increase the debris thickness of debris-coverered tongues of 'normal' valley glaciers. I suggest addressing this point more clearly in the manuscript, since the increase in debris supply and increase in supraglacial debris covered area and debris thickness are a major part of the proposed Paraglacial Transition pathway and also play a crucial role for point 2 above (glacier-rock glacier continuum).

This is an important debate.  Our view is that as glaciers retreat to form ice remnants in high cirque basins as the climate warms, a couple things happen. The glacier slows down dynamically; it thins, shrinks in area, slows its movement, and slows its erosion of the bed. In addition, the area of cirque headwalls adjacent to the glacier increases (at least increases relative to the perimeter length and area of the glacier) and this increases the area of the backwall that is capable of failure, driving the accumulation of debris to the glacier surface.

As a result, debris delivery increases to the remnant glacier by mass wasting (rock topples and small landslides), and new rock glaciers (periglacial) will develop. So, basal erosion decreases, periglacial and paraglacial rock delivery increases. The upper part of the cirque glacier generally will not have lateral moraines, because it is in the accumulation zone, and it may well be convex, and debris can reach it. But snow and ice avalanches onto the glacier can add rock debris (as well as ice/snow) to the glacier.

We argue that with sustained glacier retreat, the paraglacial and periglacial component of rock debris to the waning glacier system will increase in relative terms; and as glaciers waste, they erode less, and with mass wasting from rock walls increasing, there should be an increasing paraglacial/periglacial component to the glacier sediment production and transport and to the internal constitution of the glacier.  So in terms of debris and ice accumulation, we argue that there is a transition from a clear glacial system to what looks increasingly like a paraglacial and periglacial system.

- Transition into rock glaciers

Many studies have disussed the presence or absence of a transition of glaciers into rock glaciers and divided the community in either the continuum- or the permafrost creep school (e.g. Berthling, 2011), a difference that comes mainly from focusing on either the genesis or morphogoly of rock glaciers. To my understanding a transition of a glacier into a rock glacier is a rather rare case, and I can not see the continuum of a typical downwasted, concave Himalayan debris-cover tongue into a rock glacier to be a common transition. Also, the presence of underlying permafrost, a requirement for the presence of rock glaciers, might also not be valid in many landscapes where currently debris-covered glacier tongues are present. Since debris-covered glaciers are rather downwasting than retreating as stated by the authors, I see the possibility of a glacier-rock glacier transition as rather small in most

places in the Himalayas, unless maybe in cases where the debris-coverd tongue ends up in a small cirque located at high altitude in permafrost conditions. The authors state the uncertainty associated to this transition (e.g. 417-418) and very long time scales needed for this to happen (e.g. l. 311), but the very low probability of this transition should be stated more clearly in the text and should therefore not be menitioned in the abstract, as it might concern only a very small subsample of Himalayan glaciers, in my opinion. Instead of explicitly mentioning rock glaciers in this manuscript I strongly suggest stating e.g. 'ice-debris landforms' instead, when talking about stagnating dead ice bodies buried under a large amount of debris, what the PT view is essentially suggesting.

Thank you for this discussion. We argue that the glacier-rock-glacier transition has now been studied quite widely (see references below) and we think that a similar large-scale transition is likely in many parts of HMA. We have added these to the text

Johnson, P.G., 1980. Glacier-rock glacier transition in the southwest Yukon Territory, Canada. *Arctic and Alpine Research*, *12*(2), pp.195-204.

Anderson, R.S., Anderson, L.S., Armstrong, W.H., Rossi, M.W. and Crump, S.E., 2018. Glaciation of alpine valleys: The glacier–debris-covered glacier–rock glacier continuum. *Geomorphology*, *311*, pp.127-142.

Monnier, S. and Kinnard, C., 2017. Pluri-decadal (1955–2014) evolution of glacier–rock glacier transitional landforms in the central Andes of Chile (30–33 S). *Earth Surface Dynamics*, *5*(3), pp.493-509.

Knight, J., Harrison, S. and Jones, D.B., 2019. Rock glaciers and the geomorphological evolution of deglacierizing mountains. *Geomorphology*, *324*, pp.14-24.

Monnier, S. and Kinnard, C., 2015. Reconsidering the glacier to rock glacier transformation problem: New insights from the central Andes of Chile. *Geomorphology*, *238*, pp.47-55.

We agree with your suggestion re 'ice-debris landforms' and have explicitly referred to this in the discussion sections (lines 407, 419, 465).

We have added reference to Jarman, D., Wilson, P. and Harrison, S., 2013. Are there any relict rock glaciers in the British mountains?. *Journal of Quaternary Science*, *28*(2), pp.131-143, which specifically discusses ice-debris landforms.

**Minor points**

-Thoughout the manuscript I was not sure if the authors talk about the Himalayas as a specific region of HMA or about HMA in general, targeting the entire arc. Please specify more clearly in the text. We have done this by making it clear we are talking about HMA.

-l. 135: "high glacier volume loss" - this contradicts partly your previous sentence (l. 132-134), in which relatively small ice volume losses were stated. We have addressed this.

-l. 140-141: This sentence is thematically disconnected to the previous one and I suggest to add some more content here to clarify. We have done this by deleting 'For instance'.

Fig. 1: It would be helpful for understanding and discussion to have also ice mass loss as an additional line in subfigures b, d and e, if possible. We have added this.

-l. 178: "unlike the approach taken by climate models" – what do you mean here? Please clearify. We agree that this phrase is ambiguous and have deleted it.

-l. 218-219: "and the slow melting of clean ice and debris-covered glaciers" – what do you mean here? This is somehow disconnected to the rest of the sentence. We have clarified this statement.

-l. 230-236: Although you mention later that a combination of both responses (MIL and PT) might exist (l. 415-417), I suggest stating the likely coexistence of both views more prominently and step back partly form the impression that either one or the other is the only view (e.g. in the abstract, discussion, conclusion). Agree

-l. 261: "ice glaciers" – do you mean "clean ice glaciers"? Done

-l; 269-270: supraglacial ice cliffs should be mentioned in line with supraglacial ponds here as melt hot spots, and appropriate literature cited. We have done this and cited Sakai, A., Nakawo, M. and Fujita, K., 1998. Melt rate of ice cliffs on the Lirung Glacier, Nepal Himalayas, 1996. *Bull. Glacier Res*, *16*(57-66) as an example.

-l. 367: compiled Done.

-l. 382, 389: please clearify the meaning of "MIS 2" and "MIS 3", as the explanation in brackets (l. 383) is not clear to me. Done.

-l. 419: "above" – do you mean "below" instead? Yes! Done.

-l. 471: "increase snowfall" – is this predicted? Yes, at high elevations. Given our understanding of increased atmospheric water vapour and climate warming.

-l. 493-494: Point 4 is unclear to me, please clearify. Agree. We have reworded this.

**References:**

Berthling, I. (2011). Beyond confusion: Rock glaciers as cryo-conditioned landforms. Geomorphology, 131(3-4), 98-106.  We have added this reference.

---

## Author Response (AR1)

21 May 2025

**Re: egusphere-2024-4033**
**Title: Will landscape responses reduce glacier sensitivity to climate change in High Mountain Asia?**

Dear Dr Delaney

Thank you for your recent suggestions and comments on our paper. We have now added a paragraph which discusses the contested issue of rock glaciers and ice-debris landforms. The paragraph is below and is now in lines 325-334.

"We can see then that the rock glacier response to deglaciation envisioned by the PT scenario is likely to be highly complex, with the full suite of rock glaciers ('periglacial' and 'glacier-derived') and other ice-debris landforms developing in different locations, regions and over different timescales. This complex response will be driven by climate change as a first order control, and debris supply and glaciological factors as secondary factors. Separate from rock glaciers, the evolution of undifferentiated ice-debris landforms during glacier recession has hardly been discussed in the cryosphere literature from HMA (although see Bolch et al. (2019). As a result, there is uncertainty in evaluating precisely how the PT scenario would develop, how quickly and what form the equilibrium landscape might present".

We hope that our paper is now suitable for publication.

Best wishes,

Stephan (on behalf of the authors)

Stephan Harrison

stephan.harrison@exeter.ac.uk

---

## Editor Decision (ED1)

**Will landscape responses reduce glacier sensitivity to climate change in High Mountain Asia?**

**Specific comments**

- Ln 73: Double check this and provide a citation.

- Figure 1: Clarify in caption that sediment supply is to the glacier, not from the glacier. Also, replace "Ice glacier" with "Clean ice glacier" as this is what is used in text. Reviewer 2 made a comment about this.

- Paragraph 169–189: Uses "explore" three times. Please change.

- Glacier lake outburst floods, in my understanding, is not capitalized.

- Ln 222: Something strange in this sentence. It seems to suggest that negative mass balance refers to lakes, not glaciers. Please rephrase.

- Ln 230: I could be wrong as a non-expert, but my impression is that "modeling research" or "regional research" could be a better representation of previous work mentioned above. Also is there a chance that a review article on this research could be cited here?

- Ln 241: Could "water pressures" be explained? The connection is unclear to me.

- Ln 261: Please add citations for the glacierMIP work?

- Ln 265: Please define "these features."

- Ln 274: "PT scenario will likely increase the resilience..." The increase happens regardless of our MIL or PT scenario. Would a better way to phrase this sentence be: "PT scenario accounts for increased resilience..."?

- Ln 280: As no citation is given at the end of the sentence, please change the "will" to a "could".

- Ln 311: is "although" needed? Same for Ln 331?

- Ln 316: "these glaciers?? move into high-elevation cirques"? "Conditions" sounds a bit strange.

- Ln 312: Should ice-debris landforms be included/mentioned here?

- Ln 320: is "more rapidly" meant rather than greater? It is not clear what exactly is greater.

- Ln 321: "rock glaciers to develop" would "transition to occur" be more accurate?

- Ln 332: "precisely how". Could this be replaced with a more concrete example, for instance, it is difficult to examine the precise timing/variability amongst glaciers and/or regions? As it is, I find that this sentence may undermine much of the message in the paper.

- Ln 340: Can some clarification be added here? Do the authors mean glaciers generally "the glaciers"? a particular glacier? or group of glaciers, for instance in a region?

- Ln 357: Consider removing this sentence and referencing the figures.

- Ln 362-363: While the sensitivity of the glaciers is discussed, could speculative statements be made (and supported) about the role the pathway or geomorphic impact?

- Figure 2: In the bar plots, what is 7.3? please add a scale to mention what they represent?

- Figure 2: LIA is noted in the figure, however, "Holocene Neoglacial Maximum" (Ln 347) or "Neoglacial" (395) are used in the text. Please make consistent.

- Figure 2: Along the recommendations of Reviewer 1, please include the Karakoram on the map, as this region is discussed in lines 408-409. Likewise, as the Khumbu region is referenced in paragraph at 418, please mention which region that it is in.

- Ln 404: MIS defined multiple times (see above). Also, would it make sense to add dates? and use either LGM or Last Glacier Maximum, but not both.

- Paragraph 399-414: This paragraph would be strengthened if a stronger link was made with the different pathways and the role of geomorphic activity.

- Ln 432-433: "supports the PT scenario..." As an outsider to the field, could a more detailed explanation be made as to why this behavior supports the PT pathway, as opposed to other factors such as glacier response time.

- Ln 485: given the specific findings (90% loss by 2100), please add a citation.

- Ln 500: "More research..." can this sentence be shortened?

- Ln 501–504: There has been some drilling in Khumbu to these ends, I believe. Please add a citation.

- Table 1. Remove blank column and consider added a column with average of the different studies to easily differentiate the differences amongst the regions.

---

## Author Response (AR2)

Dear Dr Delaney

Thank you for your further suggestions. Below you will see our responses (in red) in response to your suggestions and these are also shown as Track Changes on the revised document.

We hope that this is the last set of revisions. The paper was submitted a long time ago and has already undergone review by two reviewers (and Reviewer 1 made substantial comments on the wording).

Best wishes,
Stephan

**Will landscape responses reduce glacier sensitivity to climate change in**

**High Mountain Asia?**

Specific comments

• Ln 73: Double check this and provide a citation. Done

• Figure 1: Clarify in caption that sediment supply is to the glacier, not from the glacier. Done

Also, replace "Ice glacier" with "Clean ice glacier" as this is what is used in text. Reviewer 2 made a comment about this. Done.

• Paragraph 169–189: Uses "explore" three times. Please change. Done.

• Glacier lake outburst floods, in my understanding, is not capitalized. Done.

• Ln 222: Something strange in this sentence. It seems to suggest that negative mass balance refers to lakes, not glaciers. Please rephrase. Agree and done

• Ln 230: I could be wrong as a non-expert, but my impression is that "modeling research" or "regional research" could be a better representation of previous work mentioned above. Also is there a chance that a review article on this research could be cited here? We have discussed this and would like to maintain the wording. There is no wider review article we think could be used here. We have cited many of the relevant papers.

• Ln 241: Could "water pressures" be explained? The connection is unclear to me. Yes, we have done this.

• Ln 261: Please add citations for the glacierMIP work? Added.

• Ln 265: Please define "these features." This relates to the previous sentence about rock glaciers. We hope this is now clear.

• Ln 274: "PT scenario will likely increase the resilience…" The increase happens regardless of our MIL or PT scenario. Would a better way to phrase this sentence be: "

PT scenario accounts for increased resilience. . . "? We disagree with this suggestion and would like to keep the text as it is.

• Ln 280: As no citation is given at the end of the sentence, please change the "will" to a "could". Agree. Done

• Ln 311: is "although" needed? Same for Ln 331? Thanks.  We use this to suggest that while the statement is correct, there are exceptions to this.

• Ln 316: "these glaciers?? move into high-elevation cirques"? "Conditions" sounds a bit strange.  Yes, agree.  We have reworded this.

• Ln 312: Should ice-debris landforms be included/mentioned here? Yes, agree.  We have done this.

• Ln 320: is "more rapidly" meant rather than greater? It is not clear what exactly is greater.  The scale of the transition. We think this is now clear.

• Ln 321: "rock glaciers to develop" would "transition to occur" be more accurate?  We disagree as the previous phrase is widely used (e.g. in Jarman et al., Ballantyne 2023).

• Ln 332: "precisely how". Could this be replaced with a more concrete example, for instance, it is difficult to examine the precise timing/variability amongst glaciers and/or regions.  As it is, I find that this sentence may undermine much of the message in the paper.  We point out that there remains much research to be carried out if we are able to assess the nature and timing of the PT view.

• Ln 340: Can some clarification be added here? Do the authors mean glaciers generally "the glaciers"? a particular glacier? or group of glaciers, for instance in a region? Glaciers generally.  We have omitted 'the'.

• Ln 357: Consider removing this sentence and referencing the figures. We have discussed this and would want to keep the sentence.

• Ln 362-363: While the sensitivity of the glaciers is discussed, could speculative statements be made (and supported) about the role the pathway or geomorphic impact? We don't want to add any more speculation (this paper already contains much of this) so have kept this.

• Figure 2: In the bar plots, what is 7.3? please add a scale to mention what they represent?  7.3 represents the average reduction in glacier length for the whole Himalaya, but was not discussed in the figure caption.  We have clarified this by using a vertical horizontal scale showing 7.3km glacier length reduction.

• Figure 2: LIA is noted in the figure, however, "Holocene Neoglacial Maximum" (Ln 347) or "Neoglacial" (395) are used in the text. Please make consistent.  This is discussed and clarified in the text.

• Figure 2: Along the recommendations of Reviewer 1, please include the Karakoram on the map, as this region is discussed in lines 408-409. Likewise, as the Khumbu region is

referenced in paragraph at 418, please mention which region that it is in. We have done these and also added 'Khumbu' to the map.

• Ln 404: MIS defined multiple times (see above). Also, would it make sense to add dates? and use either LGM or Last Glacier Maximum, but not both. We have clarified this.

• Paragraph 399-414: This paragraph would be strengthened if a stronger link was made with the different pathways and the role of geomorphic activity. As this is speculative, we don't think this link is warranted.

• Ln 432-433: "supports the PT scenario…" As an outsider to the field, could a more detailed explanation be made as to why this behavior supports the PT pathway, as opposed to other factors such as glacier response time. Downwasting is a clear response of Himalayan glaciers to negative mass balance, and we argue this supports the PT pathway.

• Ln 485: given the specific findings (90% loss by 2100), please add a citation. Done.

• Ln 500: "More research…" can this sentence be shortened? This is only two lines in length, so we are not sure why this needs to be shortened.

• Ln 501–504: There has been some drilling in Khumbu to these ends, I believe. Please add a citation. This work by Duncan Quincey and colleagues was not directed at our research questions. They wanted to assess the thermal regime of the Khumbu glacier and did not discuss the transition to rock glaciers.

• Table 1. Remove blank column and consider added a column with average of the different studies to easily differentiate the differences amongst the regions. We do not agree that adding an averaged value would add to this argument.

Thanks for your suggestions.